

# Space borne tropospheric nitrogen dioxide (NO₂) observations from 2005-2020 over the Yangtze River Delta (YRD), China: variabilities, implications, and drivers

Hao Yin [1, 2], Youwen Sun [1,†], Justus Notholt[3], Mathias Palm[3], and Cheng Liu[2,4,5,6]

[1]*Key Laboratory of Environmental Optics and Technology, Anhui Institute of Optics and Fine Mechanics, HFIPS, Chinese Academy of Sciences, Hefei 230031, China*

[2] *Department of Precision Machinery and Precision Instrumentation, University of Science and Technology of China, Hefei 230026, China*

[3] *University of Bremen, Institute of Environmental Physics, P. O. Box 330440, 28334 Bremen, Germany*

[4] *Anhui Province Key Laboratory of Polar Environment and Global Change, University of Science and Technology of China, Hefei 230026, China*

[5] *Center for Excellence in Regional Atmospheric Environment, Institute of Urban Environment, Chinese Academy of Sciences, Xiamen 361021, China*

[6] *Key Laboratory of Precision Scientific Instrumentation of Anhui Higher Education Institutes, University of Science and Technology of China, Hefei 230026, China*

[†]Correspondence to: Youwen Sun (ywsun@aiofm.ac.cn)

## Abstract

Nitrogen dioxide (NO₂) is mainly affected by local emission and meteorology rather than long-range transport. Accurate acknowledge of its long-term variabilities and drivers are significant for understanding the evolutions of economic and social development, anthropogenic emission, and the effectiveness of pollution control measures on regional scale. In this study, we quantity the long-term variabilities and the underlying drivers of NO₂ from 2005 to 2020 over the Yangtze River Delta (YRD), one of the most densely populated and highly industrialized city clusters in China, using OMI space borne observations and the multiple linear regression (MLR) model. We have compared the space borne tropospheric results to the surface in-situ data, yielding correlation coefficients of 0.8 to 0.9 over all megacities within the YRD. As a result, the tropospheric NO₂ column measurements can be used as representatives of near-surface conditions, and we thus only use ground-level meteorological data for MLR regression. The inter-annual variabilities of tropospheric NO₂ vertical column densities (VCDs) from 2005 to 2020 over the YRD can be divided into two stages. The first stage was from 2005 to 2011, which showed overall increasing trends with a wide range of $(1.91 \pm 1.50)$ to $(6.70 \pm 0.10) \times 10^{14}$ molecules/cm²·yr⁻¹ (p<0.01) over the YRD. The second stage was from 2011 to 2020, which showed over all decreasing trends of $(-6.31 \pm 0.71)$ to $(-11.01 \pm 0.90) \times 10^{14}$ molecules/cm²·yr⁻¹ (p<0.01) over each of the megacities. The seasonal cycles of tropospheric NO₂ VCDs over the YRD are mainly driven by meteorology (81.01% - 83.91%) except during winter when anthropogenic emission contributions are pronounced (16.09% - 18.99%). The inter annual variabilities of tropospheric NO₂ VCDs are mainly driven by anthropogenic emission (69.18% - 81.34%) except for a few years such as 2018 which are partly attributed to meteorology anomalies (39.07% - 91.51%). The increasing trends in tropospheric NO₂ VCDs from 2005 to 2011 over the YRD are mainly attributed to high energy consumption associated with rapid economic growth which cause significant increases in anthropogenic NO₂ emissions. The decreasing trends in



tropospheric NO$_2$ VCDs from 2011 to 2020 over the YRD are mainly attributed to the stringent clean air measures which either adjust high energy industrial structure toward low energy industrial structure or directly reduce pollutant emissions from different industrial sectors.

Keywords: OMI; nitrogen dioxide; Emissions; Meteorology; Multiple linear regression model

## 1. Introduction

As a major tropospheric pollutant, nitrogen dioxide (NO$_2$) not only threatens human health and crop growth but also involves in a series of atmospheric photochemical reactions (Yin et al., 2019;Wang et al., 2011;Geddes et al., 2012). NO$_2$ is a crucial precursor in the formation of ozone (O$_3$), particulate matter (PM), acid rain, and photochemical smog in the troposphere (Yin et al., 2021a;Lu et al., 2019a;Lu et al., 2019b;Sun et al., 2018). Since severe NO$_2$ pollution increases the risk of respiratory disease and is highly associated with mortality (Meng et al., 2021;MacIntyre et al., 2014;Tao et al., 2012), many countries take the NO$_2$ level as an important pollution indicator of air quality (Xue et al., 2020). The sources of tropospheric NO$_2$ are mainly from anthropogenic emissions through high temperature combustions, like transportation (vehicles, ships, and airplanes) and industrial facilities (petrochemicals and power plants) (Zheng et al., 2018b;Chi et al., 2021;van Geffen et al., 2015). Additional minor sources of NO$_2$ are attributed to natural emissions from biogeochemical reaction in soil, volcanic eruption, and lightning (Bond et al., 2001;Zhang et al., 2003;Lu et al., 2021). The dominant sink of tropospheric NO$_2$ is attributed to a chemical destruction which first converts NO$_2$ into nitric acid (HNO$_3$) and peroxyacetyl nitrate (PAN) which then are by dry or wet deposition (Browne et al., 2013). Due to a short lifetime of a few hours, tropospheric NO$_2$ is heavily affected by local emissions and meteorology rather than long-range transport (Kim et al., 2015;Cheng et al., 2012).

Many scientists have used a suite of active and passive observation technologies onboard ground-based, vehicle-based, ship-based, airborne, or space borne platforms to assess the temporal-spatial variabilities of NO$_2$ and identify their driving forces in different regions around the globe (Richter et al., 2005;Jiang et al., 2018;Liu et al., 2018;Zhang et al., 2021;Schreier et al., 2015;Shaiganfar et al., 2017). Among all observation technologies and platforms, space borne remote sensing observations have their unique features. By validating with ground-based remote sensing or balloon observations, space borne observations can provide global NO$_2$ dataset with a reasonable accuracy. Typical space borne instruments include the SCIAMACHY, GOME, OMI, and TROPOMI, which have been widely used in scientific investigations of global nitrogen cycle, O$_3$ formation regime, and regional pollution & transport, quantification of NO$_2$ emissions from biomass burning regions, megacities, and industrial facilities, and validation of shipborne observations and atmospheric chemical transport models (CTMs) (Richter et al., 2005;Bechle et al., 2013;Boersma et al., 2011;Ghude et al., 2009;Lamsal et al., 2008). Using space borne observations to derive long term trends of NO$_2$ and their drivers not only provides valuable information for evaluation of regional emissions, but also improves our understanding of atmospheric evolutions. Richter et al., (2005) first investigated the inter annual variabilities of tropospheric NO$_2$ vertical column densities (VCDs) from space with GOME and SCIAMACHY observations during 1996-2004. Richter et al., (2005) found substantial reductions in NO$_2$ VCDs over some areas of Europe and the USA, but a highly significant increase of about 50%—with an accelerating trend in annual growth rate—over the industrial areas of China. In a subsequent study, Ghude et al., (2009) found the same





phenomenon as those of Richter et al., (2005) with GOME and SCIAMACHY observations from
1996 to 2006, which disclosed that tropospheric $NO_2$ VCDs showed increasing trends over the rapid
developing regions (China: 11 ± 2.6%/year, South Asia: 1.76 ± 1.1%/year, Middle East Africa: 2.3
± 1 %/year) and decreasing or level-off trends over the developed regions (US: -2 ± 1.5%/year,
Europe: 0.9 ± 2.1%/year). With multiple satellite platforms including GOME, SCIAMACHY, OMI,
and GOME-2, Hilboll et al., (2013) also found 5% to 10% $yr^{-1}$ of increasing trends for tropospheric
$NO_2$ VCDs over eastern Asia during 1996 to 2011 . With the OMI observations, Lamsal et al., (2015)
have quantified the $NO_2$ trend from 2005 to 2013 over the US and Krotkov et al., (2016) have
investigated the $NO_2$ trends over different countries for the period of 2005–2014.
Along with the great advances in social and economic development in recent decades, air
quality in China has changed dramatically (Sun et al., 2020;Sun et al., 2021c;Yin et al., 2020;Yin et
al., 2021c;Yin et al., 2021d). China has implemented a series of clean air measures in different stages
to tackle air pollution across China. One of the landmark clean air measures could be the Action
Plan on the Prevention and Control of Air Pollution implemented in 2013, which launched many
stringent measures to improve air quality across China. These measures include the reduction of air
pollutant emissions, the adjustment of industrial structure and energy mix, the establishment of
early-warning systems and monitoring for air pollution, and other compulsive policies (China State
Council, 2013). Both space borne and ground-based observations have witnessed the effectiveness
of these successful polices. The OMI tropospheric $NO_2$ VCDs have been decreased by 21% from
2011 to 2015 over 48 cities of China (Liu et al., 2017). The national averaged surface $NO_2$ recorded
by the China National Environmental Monitoring Center (CNEMC) network has significantly
decreased from (16.68 ± 4.82) ppbv in 2013 to (11.29 ± 3.25) ppbv in 2020 (Lin et al., 2021).
In this study, we use tropospheric $NO_2$ VCDs from 2005-2020 provided by OMI to
comprehensively evaluate the long-term trends, implications, and underlying drivers of $NO_2$ over
the Yangtze River Delta (YRD, including Anhui, Jiangsu, Shanghai, and Zhejiang Provinces). In
addition to anthropogenic emission, meteorology also drives $NO_2$ variability by affecting emissions,
transport, chemical production, and scavenging. The relationships of $NO_2$ against meteorological
variables are complex and are region and time dependent. In present work, we separate the
contributions of meteorology and anthropogenic emission to the $NO_2$ variability by multiple linear
regression (MLR) model over the major cities (Hefei, Nanjing, Suzhou, Shanghai, Hangzhou,
Ningbo) within the YRD. As one of the three most densely populated and highly industrialized city
clusters in China, the YRD has long been identified as a key region for air pollution mitigation. This
study can not only improve our understanding of temporal spatial $NO_2$ evolutions in the atmosphere
but also provides valuable information for future clean air policy. We introduce detailed descriptions
of OMI and ground-level $NO_2$ products in section 2.1, and meteorological fields in section 2.2. The
method for separating contributions of meteorology and anthropogenic emission is presented in
section 2.3. Sections 3.1 and 3.2 analyze the temporal-spatial variabilities of tropospheric $NO_2$ from
2005 to 2020 over the YRD on provincial and megacity levels, respectively. A comparison between
the OMI $NO_2$ product and the ground-level measurements is performed in section 3.3. We discuss
the implications and underlying drivers of the variabilities of tropospheric $NO_2$ from 2005 to 2020
over the YRD in section 4. We conclude this study in section 5.
**2. Data and method**
**2.1 Observation data**



### 2.1.1 OMI NO$_2$ product

OMI is a hyperspectral atmospheric composition detection instrument onboard the National Aeronautics and Space Administration (NASA) Aura Earth Observing System (EOS) satellite launched in July, 2004 (Boersma et al., 2007). The EOS satellite flies over a low-Earth orbit at an altitude of about 710 km. The local overpass time (LT) of OMI satellite is about 13:30 in early afternoon. The retrieval micro window for NO$_2$ VCDs lies in between 405 nm and 465 nm with a spectral resolution of about 0.5nm (Marchenko et al., 2015). The spatial resolution of OMI measurements is $13 \times 24$ km$^2$ at nadir. OMI provides observations of O$_3$, NO$_2$, SO$_2$, aerosol, cloud, HCHO, BrO, and OClO with nearly daily global coverage (Levelt et al., 2006). The daily LV3 data product of tropospheric NO$_2$ VCDs data (GES DISC; http://disc.sci.gsfc.nasa.gov, last accessed: 1 September 2021) which is a gridded data with a $0.25° \times 0.25°$ spatial resolution are used in this study. The tropospheric NO$_2$ VCDs are calculated by Stratosphere–troposphere separation (STS) scheme proposed by numerous previous studies (Bucsela et al., 2013;Lamsal et al., 2014;Goldberg et al., 2017). The STS scheme first subtract the stratospheric NO$_2$ slant column densities (SCDs) from the total NO$_2$ SCDs and then it divides the resulting tropospheric NO$_2$ SCDs by the tropospheric air mass factor (AMF). The formulation for calculating tropospheric NO$_2$ VCDs is as follow:

$$VCD_{trop} = \frac{SCD_{total} - SCD_{strat}}{AMF_{trop}} \tag{1}$$

where AMF is defined as the ratio of the SCD to the VCD (Solomon et al., 1987),

$$AMF_{trop} = \frac{SCD_{trop}}{VCD_{trop}} \tag{2}$$

The tropospheric AMF are calculated by NO$_2$ profiles simulated by the Global Modeling Initiative (GMI) chemistry transport model with the horizontal resolution of $1° \times 1.25°$ (Rotman et al., 2001). Separation of stratospheric and tropospheric columns is achieved by the local analysis of the stratospheric field over unpolluted areas (Bucsela et al., 2013). The OMI tropospheric NO$_2$ VCDs dataset has been used in many studies to investigate O$_3$ formation regime and regional pollution & transport (Lin et al., 2010;Zhang et al., 2017;Duncan et al., 2013;Liu et al., 2016). In this study, only the LV3 data product collected with cloud radiance fractions of less than 30% is used (Streets et al., 2013).

### 2.1.2 Ground level NO$_2$ data

We extract ground level NO$_2$ data over the YRD from the China National Environmental Monitoring Center (CNEMC) network (http://www.cnemc.cn/en/, last access: November 26, 2021). The CNEMC network has operated more than 3000 monitoring sites that almost cover all major cities over China by 2020. The CNEMC datasets have been used in many studies for evaluation of regional atmospheric pollution & transport (Li et al., 2021;Lu et al., 2019a;Lu et al., 2020;Sun et al., 2021a;Yin et al., 2021;Zhao et al., 2016;He et al., 2017). As one of the six key atmospheric pollutants (CO, SO$_2$, NO$_2$, PM$_{10}$, O$_3$, and PM$_{2.5}$) routinely measured by the CNEMC network, ground level NO$_2$ measurements at 188 sites in 40 cities over the YRD are available since 2014. In this study, comparisons between the OMI NO$_2$ data product and the ground level NO$_2$ measurements are only performed over 6 key megacities, i.e., Shanghai, Nanjing, Hangzhou, Suzhou, Ningbo, and Hefei, within the YRD. The population, geolocation, the number of measurement site, and data



range of each city are summarized in Table 1. The number of measurement site in each city ranges
from 8 to 11, the altitude ranges from 3 to 50 m (above sea level, a.s.l.), and the population ranges
from 0.9 to 2.5 million. All ground level $NO_2$ data at each station are measured by active differential
absorption ultraviolet (UV) analyzers. We use a data quality control method following previous
studies to remove unreliable $NO_2$ data (Lu et al., 2019a;Lu et al., 2020;Sun et al., 2021a;Yin et al.,
2021a). Specifically, we first convert all hourly measurements into $Z$ scores, we then remove the
measurement if its $Z$ score meets one of the following rules: (1) $Z_i$ is larger or smaller than the
previous value $Z_{i-1}$ by 9 ($|Z_i - Z_{i-1}| > 9$); (2) The absolute value of $Z_i$ is greater than 4 ($|Z_i| >$
4); (3) the ratio of the $Z$ value to the third-order center moving average is greater than 2 ($\frac{3Z_i}{Z_{i-1}+Z_i+Z_{i+1}} >$
2), where $i$ represents the $i^{th}$ hourly measurement data. After removing OUTLIERS with above filter
criteria, we finally average $NO_2$ data at all measurement sites in each city to form a city
representative $NO_2$ dataset.

**2.2 Meteorological fields**

We obtain meteorological fields during 2005-2020 from the second Modern-Era Retrospective
analysis for Research and Applications (MERRA-2) (Gelaro et al., 2017). This dataset is produced
by the NASA Global Modeling and Assimilation Office
(https://gmao.gsfc.nasa.gov/reanalysis/MERRA-2/, last accessed: 1 August, 2021) with a spatial
resolution of 0.5° × 0.625°, temporal resolutions of 1 h for boundary layer height and surface
meteorological variables, and 3 h for other variables. Previous studies have verified that
meteorological fields provided by MERRA-2 match well with the meteorological parameters
observed by Chinese weather stations (Song et al., 2018;Carvalho, 2019;Wang et al., 2017;Kishore
Kumar et al., 2015;Zhou et al., 2017). In order to match OMI observations which are available at
about 13:30 LT, the average for meteorological data is only performed between 13:00 and 14:00 LT.

**2.3 Multiple linear regression (MLR) model**

We establish a multiple linear regression (MLR) model to quantify the contributions of
meteorology and anthropogenic emission to the long-term variabilities of tropospheric $NO_2$ VCDs
during 2005-2020 over the YRD. Similar MLR methodologies have been used in previous studies
to estimate the contributions of meteorology and emission to the variabilities of $O_3$ and $PM_{2.5}$ in
North America, Europe and China (Li et al., 2019;Li et al., 2020;Xu et al., 2011;Zhai et al.,
2019;Zhao and Wang, 2017). The meteorological parameters used in our MLR model are elaborated
in Table 2.
In order to highlight the variabilities of tropospheric $NO_2$ VCDs, we follow the method of
previous studies and calculate tropospheric $NO_2$ VCDs anomalies ($\boldsymbol{y}_{anomaly}$) by subtracting a
reference value ($\boldsymbol{y}_{reference}$) from all tropospheric $NO_2$ observations ($\boldsymbol{y}_{individual}$) (Hakkarainen et
al., 2016;Hakkarainen et al., 2019;Mustafa et al., 2021). The formulation of this method is expressed
as:

$$\boldsymbol{y}_{anomaly} = \boldsymbol{y}_{individual} - \boldsymbol{y}_{reference} \qquad (3)$$

In this study, we take the average of all tropospheric $NO_2$ VCDs from 2005 to 2020 (i.e., the
16-year mean) as the reference value. The MLR model for each city is explained as:


$$y = \beta_0 + \sum_{k=1}^{11} \beta_k x_k \qquad (4)$$
where $y$ are the regression result for monthly OMI tropospheric $NO_2$ VCDs anomalies, $\beta_0$ is
the intercept, and $x_k$ ($k \in [1, 11]$) are the meteorological variables. The regression coefficients $\beta_k$
are calculated by nonlinear least squares fitting. This MLR model finds the optimal regression result
by minimizing the sum of squares of the fitting residual and then solves regression coefficients $\beta_k$
by the following equation:
$$\beta_k = (\sum x_k x_k^T)^{-1}(\sum x_k y_k) \qquad (5)$$
The regression results $y$ represent the meteorology induced contributions to the variabilities
of tropospheric $NO_2$ VCDs. Since both soil and lighting $NO_x$ are meteorology dependent, the effects
of soil and lighting $NO_x$ on $NO_2$ variability are also attributed to meteorology contribution. The
difference $y'$ between the monthly OMI tropospheric $NO_2$ VCDs anomalies $y_{anomaly}$ and $y$
calculated as equation (6) represents the portion that cannot be explicitly explained by the
meteorological influence.
$$y' = y_{anomaly} - y \qquad (6)$$
By subtracting the meteorological influence from the total $NO_2$ amounts, the $y'$ is referred to
as the aggregate contribution of anthropogenic emission. Positive $y$ and $y'$ indicate that
meteorology and anthropogenic emission cause tropospheric $NO_2$ VCDs above the reference value
(i.e., the 16-year mean), respectively. In contrast, negative $y$ and $y'$ indicate that meteorology and
anthropogenic emission cause tropospheric $NO_2$ VCDs below the reference value, respectively.
Since the meteorological parameters listed in Table 2 differ in units and magnitudes, which
could lead to unstable performance of the model. Therefore, we normalized all meteorological
parameters via equation (7) before using them in regression. This normalization pre-processing
procedure can also speed up the convergence of the MLR model.
$$z_k = \frac{x_k - u_k}{\sigma_k} \qquad (7)$$
where $u_k$ and $\sigma_k$ are the average and 1σ standard deviation (STD) of $x_k$, and $z_k$ is the
normalized value for parameter $x_k$.
**3. Temporal-spatial variabilities of tropospheric $NO_2$ VCDs over the Yangtze River Delta**
**3.1 Variabilities at provincial level**
We present the temporal-spatial distribution of the annual averaged tropospheric $NO_2$ VCDs
over the YRD from 2005 to 2020 in Figure 1. The major pollution areas for tropospheric $NO_2$ VCDs
over the YRD are located in the south of Jiangsu Province and north of Zhejiang Province. In
addition, $NO_2$ pollution in eastern Anhui Province showed an increasing trend during 2005-2013
and became one of the major pollution areas within YRD during 2010-2013. The amplitudes of
tropospheric $NO_2$ VCDs over the YRD showed large year to year variabilities from 2005 to 2020
but spatial extensions of the major pollution areas are almost constant over years. Among all the
pollution areas, the heaviest pollution regions are uniformly located in the densely populated and
highly industrialized megacities such as Shanghai, Nanjing, Suzhou, Hangzhou, Ningbo, and Hefei.
The annual means and seasonal cycles of tropospheric $NO_2$ VCDs over the YRD during 2005-
2020 at Province or municipality level, i.e., Anhui Province, Jiangsu Province, Zhejiang Province,
and Shanghai municipality, are presented in Figure 2. Tropospheric $NO_2$ VCDs over each province
are calculated by averaging all observations within the boundary of each province. For seasonal





variability, clear seasonal features over the whole YRD region and each province are observed
(Figure 2a): (1) high levels of tropospheric $NO_2$ VCDs occur in late winter to spring and low levels
of tropospheric $NO_2$ VCDs occur in later summer to autumn; (2) the $1\sigma$ STDs in late winter to spring
are larger than those in later summer to autumn; and (3) seasonal cycles of tropospheric $NO_2$ VCDs
over Jiangsu, Zhejiang and the whole YRD region show bimodal patterns, i.e., two seasonal peaks
occur around March and December or January, and one seasonal trough occurs around September;
but these over Anhui shows a unimodal pattern and don't have the peak around March. The
tropospheric $NO_2$ VCDs present a maximum monthly mean value of $(1.93 \pm 0.21)$, $(2.40 \pm 0.25)$,
$(1.61 \pm 0.16)$, and $(1.91 \pm 0.16) \times 10^{16}$ molecules/$cm^2$ in January or December over Anhui, Jiangsu,
Zhejiang, and the whole YRD region, respectively. The minimum monthly mean values over Anhui,
Jiangsu, Zhejiang and the whole YRD region occur in July, with values of $(0.35 \pm 0.05)$, $(0.83 \pm$
$0.07)$, $(0.57 \pm 0.06)$, and $(0.39 \pm 0.01) \times 10^{16}$ molecules/$cm^2$, respectively.

Except a few anomalies such as the year-to-year decrease in 2005-2006, and the increases in

2016-2017 and 2018-2019, the overall inter annual variabilities of tropospheric $NO_2$ VCDs over the
YRD can be divided into two stages (Fig. 2b). The first stage was from 2005 to 2011, which showed
overall increasing trends in tropospheric $NO_2$ VCDs over the YRD. During 2005 to 2009 of this
stage, change rates of tropospheric $NO_2$ VCDs were less pronounced, where the 2009 relative to
2005 levels have only increased by $(0.33 \pm 0.02) \times 10^{15}$ $(3.96 \pm 0.25)$ %, $(1.05 \pm 0.11) \times 10^{15}$ $(8.55 \pm$
$0.08)$ %, and $(0.46 \pm 0.03) \times 10^{15}$ molecule/$m^2$ $(5.05 \pm 0.32)$ % over Anhui, Jiangsu and the whole
YRD region, respectively, and leveled off over Zhejiang. However, tropospheric $NO_2$ VCDs in 2011
relative to 2009 showed significantly increments of $(2.88 \pm 0.23) \times 10^{15}$ $(33.78 \pm 2.70)$ %, $(3.81 \pm$
$0.32) \times 10^{15}$ $(29.01 \pm 2.45)$ %, $(2.08 \pm 0.18) \times 10^{15}$ $(27.97 \pm 2.43)$ %, $(2.10 \pm 0.19) \times 10^{15}$ molecule/$m^2$
$(21.59 \pm 1.95)$ % over Anhui, Jiangsu, Zhejiang and the whole YRD region, respectively. The second
stage was from 2011 to 2020, which showed overall decreasing trends in tropospheric $NO_2$ VCDs
over the YRD. The total decrements over Anhui, Jiangsu, Zhejiang and the whole YRD region in
2020 relative to 2011 are $(4.91 \pm 0.39) \times 10^{15}$ $(41.48 \pm 3.30)$ %, $(4.82 \pm 0.31) \times 10^{15}$ $(43.25 \pm 2.72)$ %,
$(3.78 \pm 0.36) \times 10^{15}$ $(40.47 \pm 4.12)$ %, $(4.91 \pm 0.39) \times 10^{15}$ molecule/$m^2$ $(41.48 \pm 3.30)$ %, respectively.

We have followed the methodology of (Li et al., 2020)) and used the linear regression model

to estimate the inter annual trends of tropospheric $NO_2$ VCDs over the YRD (Table 3). During 2005-
2011, inter annual trends of tropospheric $NO_2$ VCDs over the YRD region and each province
spanned a wide range of $(1.74 \pm 0.72) \times 10^{14}$ molecules/$cm^2 \cdot yr^{-1}$ (p=0.02) to $(5.94 \pm$
$1.01) \times 10^{14}$ molecules/$cm^2 \cdot yr^{-1}$ (p<0.01), indicating a regional representative of each dataset. In
contrast, inter annual trends of tropospheric $NO_2$ VCDs over the YRD region and each province
from 2011 to 2020 varied over $(-4.86 \pm 0.49)$ to $(-8.16 \pm 0.82) \times 10^{14}$ molecules/$cm^2 \cdot yr^{-1}$ (p<0.01).
For the aggregate trends during 2005-2020, tropospheric $NO_2$ VCDs over the whole YRD region
and each province are negative. The largest and lowest decreasing trends are observed in Jiangsu
and Anhui, with values of $(-1.92 \pm 0.30) \times 10^{14}$ molecules/$cm^2 \cdot yr^{-1}$ (p<0.01) and $(-0.92 \pm 0.26) \times 10^{14}$
molecules/$cm^2 \cdot yr^{-1}$ (p<0.01), respectively.
**3.2 Variabilities at megacity level**

The annual means and seasonal cycles of tropospheric $NO_2$ VCDs over the major megacities

within YRD during 2005-2020 are presented in Figure 3. Similar to the derivation of provincial
level $NO_2$, tropospheric $NO_2$ VCDs over each megacity are calculated by averaging all observations



within the boundary of each megacity. The results show that the amplitudes and variabilities of
tropospheric $NO_2$ VCDs at megacity level are basically coincident with those at the corresponding
provincial levels. Overall, the amplitudes and $1\sigma$ STDs of $NO_2$ seasonal cycles in cold seasons are
larger than those in warm seasons, and the inter annual $NO_2$ variabilities at megacity level can also
be divided into two stages, i.e., an overall increasing stage during 2005-2011 and a decreasing stage
during 2011-2020. As a result, it is feasible to select these major megacities as representatives for
mapping the drivers of $NO_2$ variabilities over the YRD.
Specifically, megacity level of tropospheric $NO_2$ VCDs show seasonal maxima in December
and seasonal minima in July. Seasonal maxima over Hefei, Shanghai, Nanjing, Suzhou, Hangzhou,
and Ningbo are $(2.03 \pm 0.15)$, $(2.80 \pm 0.23)$, $(2.62 \pm 0.25)$, $(2.66 \pm 0.16)$, $(1.83 \pm 0.18)$, and $(2.27 \pm$
$0.21)\times10^{16}$ molecules/$cm^2$, and seasonal minima are $(0.34 \pm 0.04)$, $(0.83 \pm 0.11)$, $(0.58 \pm 0.06)$, $(0.62$
$\pm 0.05)$, $(0.32 \pm 0.02)$, and $(0.38 \pm 0.03)\times10^{16}$ molecules/$cm^2$, respectively. The seasonal maxima
are on average $(82.27 \pm 2.34)$ %, $(67.19 \pm 1.56)$ %, $(71.06 \pm 2.32)$ %, $(83.33 \pm 3.05)$ %, $(77.62 \pm$
$2.89)$ %, and $(70.84 \pm 2.76)$ % higher than the seasonal minima over respective megacity. As
commonly observed, the seasonal variability of tropospheric $NO_2$ VCDs with respect to their annual
means spanned a wide range of $-55.1\%$ to $103.5\%$ depending on season and measurement time
(Figure 3a).
Tropospheric $NO_2$ VCDs in all megacities show the maximum values in 2011, where the
maximum values over Hefei, Shanghai, Suzhou, Ningbo, Nanjing and Hangzhou are $(1.41 \pm 0.25)$,
$(2.18 \pm 0.23)$, $(1.81 \pm 0.17)$, $(1.39 \pm 0.12)$, $(1.88 \pm 0.18)$ and $(1.19 \pm 0.14)\times10^{16}$ molecules/$cm^2$,
respectively (Figure 3b). In terms of the increments relative to the 2005 levels, Hefei and Shanghai
from 2005 to 2011 have the largest and lowest increments of $(5.37 \pm 0.51)\times10^{15}$ molecules/$cm^2$
$(61.77 \pm 5.87)$ % and $(2.62 \pm 0.27)\times10^{15}$ molecules /$cm^2$ $(14.68 \pm 1.51)$ %, respectively. The
increments over other cities varied over $(3.31 \pm 0.32)\times10^{15}$ molecules /$cm^2$ $(31.20 \pm 3.02)$ % to $(5.21$
$\pm 0.41)\times10^{15}$ molecules/$cm^2$ $(38.40 \pm 3.02)$ %. In terms of the decrements relative to the 2011 levels,
Shanghai and Hangzhou from 2011 to 2020 have the largest and lowest decrements of $(9.77 \pm$
$0.82)\times10^{15}$ molecules/$cm^2$ $(46.89 \pm 3.94)$ and $(5.28 \pm 0.45)\times10^{15}$ molecules/$cm^2$ $(45.43 \pm 3.87)$ %,
respectively. The decrements over other cities are also evident and varied over $(6.33 \pm 0.58)\times10^{15}$
molecules/$cm^2$ $(45.53 \pm 4.18)$ % to $(9.05 \pm 0.98)\times10^{15}$ molecules/$cm^2$ $(48.12 \pm 5.21)$ %. A few
anomalies are also observed in some megacities and are in good agreement with the corresponding
provincial levels. For example, tropospheric $NO_2$ VCDs over Hefei and Suzhou had increased by
$(0.09 \pm 0.01)\times10^{15}$ molecules/$cm^2$ $(0.77 \pm 0.09)$ % and $(0.80 \pm 0.07)\times10^{15}$ molecules/$cm^2$ $(4.90 \pm$
$0.43)$ % in 2013 relative to 2012 levels, respectively. In addition, tropospheric $NO_2$ VCDs over
Hefei, Shanghai, Nanjing, Hangzhou, and Suzhou had increased by $(0.65 \pm 0.12)\times10^{15}$ $(8.41 \pm$
$1.55)$ %, $(0.35 \pm 0.02)\times10^{15}$ $(2.66 \pm 0.15)$ %, $(0.86 \pm 0.18)\times10^{15}$ $(8.16 \pm 1.71)$ %, $(0.55 \pm 0.08)\times10^{15}$
$(8.68 \pm 1.26)$ %, and $(0.29 \pm 0.05)\times10^{15}$ molecules/$cm^2$ $(2.52 \pm 0.43)$ % in 2019 relative to 2018
levels, respectively.
The inter annual trends of tropospheric $NO_2$ VCDs during 2005-2011 over all cities are positive
and span a wide range of $(1.91 \pm 1.50)$ to $(6.70 \pm 0.10)\times10^{14}$ molecules/$cm^2\cdot yr^{-1}$ $(p<0.01)$ (Table 4).
In contrast, the inter annual trends of tropospheric $NO_2$ VCDs during 2011-2020 over all cities are
negative. The largest and lowest decreasing trends are observed in Nanjing and Hangzhou, with
values of $(-11.01 \pm 0.90)$ and $(-6.31 \pm 0.71)\times10^{14}$ molecules/$cm^2\cdot yr^{-1}$ $(p<0.01)$, respectively. For the
aggregate trends during 2005-2020, tropospheric $NO_2$ VCDs over all cities are negative. The largest
and lowest decreasing trends are observed in Shanghai and Hefei, with values of $(-4.58 \pm 0.43)\times10^{14}$





molecules/cm$^2$·yr$^{-1}$ (p<0.01) and (-0.30 ± 3.43)×10$^{14}$ molecules/cm$^2$·yr$^{-1}$ (p=0.385), respectively.

### 3.3 Comparisons with the CNMEC data


In order to investigate if satellite column measurements can represent the near surface

variabilities, we have compared the OMI tropospheric NO$_2$ VCDs data over the 6 megacities within
the YRD with the ground level NO$_2$ data provided by the CNMEC (Figure 4). The comparisons over
all megacities were performed on monthly basis between June 2014 and December 2020. For each
city, the CNMEC ground level NO$_2$ measurements nearest to the OMI ground grid were included
for comparison. Considering the overpass time of OMI is at about 13:30 LT, we only average the
ground level NO$_2$ data between 13:00 and 14:00 LT for comparison, which ensures that the temporal
differences between the CNMEC and OMI dataset are all within ± 30 minutes. With these rules,
there are over 700 matching samples in each city available for comparison.

Correlation plots of OMI tropospheric NO$_2$ VCDs data against the CNMEC ground level NO$_2$

measurements are shown in Figure 4. The results show that the NO$_2$ variabilities observed by OMI
and the CNMEC are in good agreements over all megacities, with correlation coefficients ($r^2$) of
0.88, 0.81, 0.89, 0.88, 0.86 and 0.83 for Hangzhou, Hefei, Nanjing, Ningbo, Shanghai, and Suzhou,
respectively. The discrepancies between OMI and CNMEC data can be mainly attributed to their
differences in temporal-spatial resolutions. OMI averages NO$_2$ concentration at about 13:30 LT over
a large coverage due to its relatively coarse spatial resolution (Wallace and Kanaroglou, 2009;Zheng
et al., 2014). The CNMEC data represent the averaged point concentrations between 13:00 and
14:00 LT around the measurement site. NO$_2$ is a short lifetime species and characterized by large
temporal-spatial variabilities. Any temporal-spatial inhomogeneity in NO$_2$ concentration could
affect the comparison (Meng et al., 2010;Wallace and Kanaroglou, 2009). Considering above
differences, the correlations of the two datasets over all megacities are satisfactory. The tropospheric
NO$_2$ column measurements can be used as representatives of near-surface conditions. As a result,
to simplify calculations, we only use ground-level meteorological data for MLR regression.

Over polluted atmosphere, the NO$_2$ column measurements can be used as representative of

near-surface conditions because tropospheric NO$_2$ has a vertical distribution that is heavily weighted
toward the surface (Kharol et al., 2015;Zhang et al., 2017;Duncan et al., 2016;Duncan et al.,
2013;Kramer et al., 2008). Many studies have taken advantage of this favourable vertical
distribution of NO$_2$ to derive surface emissions of NO$_2$ from space (Silvern et al., 2019;Boersma et
al., 2009;Streets et al., 2013;Anand and Monks, 2017;Lu et al., 2015;Ghude et al., 2013;Cooper et
al., 2020). Meanwhile, the use of NO$_2$ column measurements to explore tropospheric O$_3$ sensitivities
has been the subject of several past studies, which disclosed that this diagnosis of O$_3$ production
rate (PO$_3$) is consistent with the findings of surface photochemistry (Jin et al., 2017;Jin and
Holloway, 2015;Sun et al., 2018;Yin et al., 2021b;Souri et al., 2017;Sun et al., 2021b;Jin et al.,
2020;Choi and Souri, 2015;Schroeder et al., 2017;Baruah et al., 2021).

### 4 Implications and drivers


We incorporate the 11 meteorological parameters listed in Table 2 into the MLR model to fit

the time series of monthly averaged tropospheric NO$_2$ VCDs from 2005 to 2020 over the 6
megacities within the YRD (Figure S1). Correlation plots of the MLR regression results and the
satellite tropospheric NO$_2$ data are shown in Figure 5. The results show that the MLR model can


well reproduce the seasonal variabilities of tropospheric $NO_2$ VCDs over each city with correlation
coefficients of 0.85 to 0.90. We separate the contributions of meteorology and anthropogenic
emission to the $NO_2$ variability over the 6 megacities with the methodology described in section 2.3.
Figure 6 shows monthly averaged tropospheric $NO_2$ VCDs along with the meteorological-driven
contributions and the anthropogenic-driven contributions in each city. Figure 7 is the same as Figure
6, but the statistics are based on annual average.

**4.1 Drivers of seasonal cycles of tropospheric $NO_2$ VCDs**

As shown in Figure 6 for all megacities, the seasonal variabilities of meteorological
contributions are consistent with those of tropospheric $NO_2$ VCDs except the period from February
to March, and the anthropogenic contributions varied around zero throughout the year except in
December and February. This means that the seasonal variabilities of tropospheric $NO_2$ over the
YRD are mainly determined by meteorology (81.01% - 83.91%) and also influenced by
anthropogenic emission in December and February. Meteorological contributions are larger than
zero in winter and lower than zero in summer, indicating that meteorology increases $NO_2$ level in
winter and decreases $NO_2$ level in summer. This contrast in meteorological contribution is associated
with the seasonal cycle of temperature. Similarly, anthropogenic contributions are larger than zero
in December and lower than zero in February, representing anthropogenic emission increases $NO_2$
level in December and decreases $NO_2$ level in February. The enhanced anthropogenic contributions
in December are mainly attributed to more extensive anthropogenic activities such as residential
heating in megacities in this period which usually results in more anthropogenic $NO_2$ emissions due
to the increase in energy and fuel consumptions. The decreased anthropogenic contributions in
February are attributed to the Spring Festival. We elaborate the analysis as bellow.
As shown in Figure S2, the vast majorities of meteorological contributions over all megacities
are from temperature and additional minor contributions over some cities such as Nanjing, Shanghai,
and Suzhou are attributed to relative humidity. Significant negative correlations between
temperature and tropospheric $NO_2$ VCDs are observed in all megacities (Figure S3, Table 5). Higher
temperature tends to decrease tropospheric $NO_2$ VCDs and vice versa. This is because higher
temperature conditions could accelerate the chemical reaction that destructs $NO_2$ in the troposphere
(Pearce et al., 2011;Yin et al., 2021a). Surface pressure and relative humidity have high positive and
negative correlations with tropospheric $NO_2$ VCDs, respectively, but their contribution levels are
much lower than the temperature. All other meteorological variables only have weak correlations
with tropospheric $NO_2$ VCDs (Table 5).
In all cities except Hefei, there is a significant increase in $NO_2$ level from February to March.
The maximum and minimum increments occur in Shanghai and Nanjing, with values of $(3.28 \pm$
$0.29) \times 10^{15}$ molecules/$cm^2$ $(16.37 \pm 1.45)$ % and $(0.47 \pm 0.05) \times 10^{15}$ molecules/$cm^2$ $(2.60 \pm 0.28)$ %,
respectively. In contrast, the meteorological contributions show decreased change rates in the same
period. As a result, this increase in $NO_2$ level from February to March could be attributed to
anthropogenic emission rather than meteorology. Indeed, anthropogenic contributions show
significant increases of $(3.95 \pm 0.32)$ to $(6.53 \pm 0.55) \times 10^{15}$ molecules/$cm^2$ over all megacities from
February to March. The most important festival in China-the Spring Festival-typically occurs in
February, when a large number of migrants in megacities return to their hometowns for holiday and
most industrial productions are shut down, which could cause significant reductions in
anthropogenic emission. In March, these migrants get back to work and all industrial enterprises



resumed productions, which could cause a rebound in anthropogenic emission. The seasonal
maxima of $NO_2$ in March are not observed in Hefei is because the anthropogenic emission induced
increases are offset by meteorology induced decreases.

2020 is a special year compared to all other years, when a large-scale lockdown occurred in

February and some regional travel restrictions occasionally occurred in other seasons across China
due to COVID-19 disease. In the comparison, we removed all $NO_2$ measurements in 2020 to
eliminate the influence of COVID-19. The monthly averaged tropospheric $NO_2$ VCDs from 2005
to 2019 along with the meteorological contributions and the anthropogenic contributions in each
city are shown in Figure S4. We obtained the same conclusion as that from Figure 6, indicating the
drivers of seasonal cycles of tropospheric $NO_2$ VCDs deduced above are consistent over years.
**4.2 Drivers of inter annual variabilities of tropospheric $NO_2$ VCDs**

As shown in Figure 7 for all megacities, the inter annual variabilities of anthropogenic

contributions are in good agreement with those of tropospheric $NO_2$ VCDs, indicating inter annual
variabilities of tropospheric $NO_2$ VCDs are mainly driven by anthropogenic emission. The same as
those of tropospheric $NO_2$ VCDs, the inter annual anthropogenic contributions over each city can
also be divided into two stages, i.e., an overall increasing stage during 2005–2011 and a decreasing
stage during 2011-2020. For the first stage (2005-2011), anthropogenic contributions account for
84.72%, 92.96%, 93.52%, 79.06%, 97.12%, and 90.21% of the increases in tropospheric $NO_2$ VCDs,
while meteorological contributions account for 15.28%, 7.04%, 6.48%, 20.94%, 2.88%, and 9.79%
over Hangzhou, Hefei, Nanjing, Ningbo, Shanghai, and Suzhou, respectively. The annual averaged
meteorological contributions over each city varied around zero in all years except few anomalies in
some years. For example, meteorological contributions over all cities are larger than zero in 2005
and 2011 but lower than zero after 2014. Pronounced anomalies include the enhancements occurred
in 2011 in all cities and the decrements in 2015 over Suzhou, in 2018 over Hangzhou, and in 2016
over other cities. All these anomalies in meteorological contributions are highly correlated with
temperature anomalies (Figure S5). As shown in Figure S6, the temperature in all cities is lower
than the reference value (i.e., the 16-year mean) in 2005 and 2011 and larger than the reference
value after 2014. As a result, in addition to anthropogenic emission, the $NO_2$ enhancements in 2011
are partly attributed to the lower temperature in this year. Meanwhile, higher temperature in YRD
region in recent years favors the decrease in tropospheric $NO_2$ VCDs. For the second stage (2011-
2020), anthropogenic contributions account for 70.15 %, 65.22 %, 66.97 %, 73.45 %, 74.43 %, and
73.84 % of the decreases in tropospheric $NO_2$ VCDs, while meteorological contributions account
for 29.85%, 34.78%, 33.03 %, 26.55 %, 25.57 %, and 26.16 % over Hangzhou, Hefei, Nanjing,
Ningbo, Shanghai, and Suzhou, respectively.

Since anthropogenic $NO_2$ emissions are highly related to economic and industrial activities

(Lin and McElroy, 2011;Russell et al., 2012;Vrekoussis et al., 2013;Guerriero et al., 2016), to
understand the inter annual variabilities of tropospheric $NO_2$ VCDs, we have investigated the inter
annual variabilities of Gross Domestic Product (GDP) over the YRD from primary sector, secondary
sector and tertiary sector (http://www.stats.gov.cn/, last accessed: 1 August, 2021) from 2005 to
2020. The primary sector includes agriculture, forestry, animal husbandry, and fishery; The
secondary industry includes mining, manufacturing, power, heat, gas and water production and
supply, and construction; The tertiary industry, namely the service industry, refers to all industries
excluded the primary industry and the secondary industry. The secondary industry is more related
to energy and fuel consumptions, and it thus dominates the anthropogenic $NO_2$ emissions. Figure
S7 shows the time series of GDP over the YRD from 2005 to 2020 and Figure 8 is the same as
Figure S7 but for year-to-year increment, i.e., the increase in GDP at a given year relative to its
previous year. The results show that the GDP of each province within the YRD increased over time
starting from 2005 but the relative contribution of each industry sector is different from year to year.
The primary sector-related GDP is relative constant, but both the secondary sector and tertiary sector
related GDPs show significant increasing trends from 2005 to 2020.
During 2009 to 2011, the GDPs have increased significantly by 198.45, 483.86, 656.40, and
327.05 billion yuan over Shanghai, Zhejiang, Jiangsu, and Anhui, where the secondary sector
contributions account for 46.50%, 53.64%, 48.99%, and 60.34% respectively. Before 2011, much
of China's economic growths still rely on the high-carbon fossil energy system and efforts to control
atmospheric pollution were relatively small. These significant increases in GDP could cause
significant increases in anthropogenic $NO_2$ emissions. After 2011, China has implemented a series
of clean air measures to tackle air pollution across China. These measures include the reduction of
industrial pollutant emissions, the adjustment of industrial structure and energy mix, and other
compulsive policies (China State Council, 2013). Zheng et al. (2018a) have estimated China's
anthropogenic emission trends from 2010 to 2017 with the bottom-up emission inventory. Zheng et
al. (2018a) found that, as the consequence of clean air measures, anthropogenic $NO_x$ emissions
across China during 2010–2017 have been decreased by 17%.
Although the total GDPs over all megacities are still increasing over time after 2011, much
of these increases are from the tertiary sector, indicating the effectiveness of the adjustment of
industrial structure and energy mix. The largest anthropogenic $NO_2$ producer from the tertiary sector
is attributed to the transportation industry including such as traffic and cargo transport, etc. Chinese
government had implemented stringent restrictions on vehicle exhaust emissions after 2011
(Ministry of Ecology and Environment of the People's Republic of China, 2016, 2011). For example,
Chinese government implemented the fourth and the fifth national motor vehicle pollutant emissions
standards in 2011 and 2018, respectively, which mandate 30% and 60% reductions in vehicle $NO_x$
emissions relative to the third national standard (Ministry of Ecology and Environment of the
People's Republic of China, 2007, 2018). These stringent measures could significantly reduce
anthropogenic $NO_2$ emissions from the tertiary sector. Overall, the decreasing trends in tropospheric
$NO_2$ VCDs from 2011 to 2020 over all megacities within the YRD are mainly attributed to the
stringent clean air measures in this period which either adjust high energy industrial structure toward
low energy industrial structure or directly reduce pollutant emissions from different industrial
sectors.
**5 Conclusions**
In this study, we have quantified the long-term variabilities and the underlying drivers of
tropospheric $NO_2$ VCDs from 2005-2020 over the Yangtze River Delta (YRD) by OMI LV3 $NO_2$
data product and MLR regressions. The major pollution areas for tropospheric $NO_2$ VCDs over the
YRD are located in the south of Jiangsu Province and north of Zhejiang Province. In addition, $NO_2$
pollution in eastern Anhui Province showed an increasing trend during 2005-2013 and became one
of the major pollution areas within YRD during 2010-2013. The amplitudes of tropospheric $NO_2$





VCDs over the YRD showed large year to year variabilities from 2005 to 2020 but spatial extensions
of the major pollution areas are almost constant over years. Among all the pollution areas, the
heaviest pollution regions are uniformly located in the densely populated and highly industrialized
megacities such as Shanghai, Nanjing, Suzhou, Hangzhou, Ningbo, and Hefei. For six megacities
the space borne tropospheric results have been compared to surface in-situ data, yielding correlation
coefficients between 0.8 and 0.9.
Clear seasonal features and inter annual variabilities of tropospheric $NO_2$ VCDs over the YRD
region are observed. Overall, the amplitudes and $1\sigma$ STDs of $NO_2$ seasonal cycles in cold seasons
are larger than those in warm seasons, and the inter annual $NO_2$ variabilities at megacity level can
be divided into two stages, i.e., an overall increasing stage during 2005-2011 and a decreasing stage
during 2011-2020. We have used the MLR regressions to quantify the drivers of tropospheric $NO_2$
VCDs from 2005 to 2020 over all megacities within the YRD. The seasonal cycles of tropospheric
$NO_2$ VCDs over the YRD are mainly driven by meteorology (81.01% - 83.91%) except in winter
when anthropogenic emission contributions are also pronounced (16.09% - 18.99%). The inter
annual variabilities of tropospheric $NO_2$ VCDs are mainly driven by anthropogenic emission (69.18%
- 81.34%) except in few years such as 2018 which are partly attributed to meteorology anomalies
(39.07% - 91.51%).

The increasing trends in tropospheric $NO_2$ VCDs from 2005 to 2011 over the YRD are mainly
attributed to high energy consumption associated with rapid economic growth which cause
significant increases in anthropogenic $NO_2$ emissions. The decreasing trends in tropospheric $NO_2$
VCDs from 2011 to 2020 over the YRD are mainly attributed to the stringent clean air measures in
this period which either adjust high energy industrial structure toward low energy industrial
structure or directly reduce pollutant emissions from different industrial sectors. This study can not
only have improved our knowledge with respect to long term evolutions of economic and social
development, anthropogenic emission, and the effectiveness of pollution control measures over the
YRD, but also have positive implications for forming future clean air policies in the important
region.
***Code and data availability.*** Surface $NO_2$ measurements over the YRD are from
http://www.cnemc.cn/en/. The OMI LV3 tropospheric $NO_2$ satellite data can be obtained from
https://acdisc.gesdisc.eosdis.nasa.gov/data/Aura_OMI_Level3/. The Chinese economic data can be
obtained from http://www.stats.gov.cn/. All other data are available on request of the corresponding
author (Youwen Sun, ywsun@aiofm.ac.cn).
***Author contributions.*** HY designed the study and wrote the paper. YS supervised and revised this
paper. JN, MP, and CL provided constructive comments.
***Competing interests.*** None.
***Acknowledgements.*** This work is jointly supported by the National Key Research and Development
Program of China (No.2019YFC0214802), the Youth Innovation Promotion Association, CAS
(No.2019434), and the Sino-German Mobility programme (M-0036).



**Table 1.** Geolocation, the number of measurement site, and population for the 6 megacities within
the YRD. Population statistics are based on the seventh nationwide population census in 2020
provided by National Bureau of Statistics of China.

| City | Latitude | Longitude | Number of sites | Altitude (m) | Population (million) |
|---|---|---|---|---|---|
| Hangzhou | 30.29 | 120.15 | 11 | 41.7 | 1.19 |
| Hefei | 31.85 | 117.25 | 10 | 29.8 | 0.94 |
| Ningbo | 29.87 | 121.55 | 9 | 5.1 | 0.94 |
| Nanjing | 32.04 | 118.77 | 9 | 8.9 | 0.93 |
| Shanghai | 31.23 | 121.47 | 10 | 4.5 | 2.49 |
| Suzhou | 31.30 | 120.62 | 8 | 3.5 | 1.28 |


**Table 2.** Meteorological parameters used in the MLR model.

| Parameters | Description |
|---|---|
| $T_{2m}$ | 2m air temperature |
| $U_{10m}$ | 10m zonal wind |
| $V_{10m}$ | 10m meridional wind |
| PBLH | Planetary boundary layer height |
| TCC | Total cloud area fraction |
| Rain | Rainfall |
| SLP | Sea level pressure |
| SWGDN | Surface incoming shortwave flux |
| $RH_{2m}$ | 2m Relative humidity |
| TROPH | Tropospheric layer Height |


**Table 3.** Inter annual trends of tropospheric $NO_2$ VCDs over each province within the YRD and the
whole YRD region during 2005 to 2011, 2011 to 2020 and 2005 to 2020.

| Province | Annual trend ($10^{14}$ molecule/m$^2$) | | |
|---|---|---|---|
| | **2005-2011** | **2011-2020** | **2005-2020** |
| YRD | 3.69 ± 0.78 (p<0.01) | -6.18 ± 0.52 (p<0.01) | -1.54 ± 0.23 (p<0.01) |
| Anhui | 4.40 ± 0.89 (p<0.01) | -5.93 ± 0.58 (p<0.01) | -0.92 ± 0.26 (p<0.01) |
| Jiangsu | 5.94 ± 1.01 (p<0.01) | -8.16 ± 0.82 (p<0.01) | -1.92 ± 0.30 (p<0.01) |
| Zhejiang | 1.74 ± 0.72 (p=0.02) | -4.86 ± 0.49 (p<0.01) | -1.41 ± 0.22 (p<0.01) |


**Table 4.** Inter annual trends of tropospheric $NO_2$ VCDs over each city within the YRD during
2005 to 2011, 2011 to 2020 and 2005 to 2020.

| Province | Annual trend ($10^{14}$ molecule/m$^2$) | | |
|---|---|---|---|
| | **2005-2011** | **2011-2020** | **2005-2020** |
| Hangzhou | 4.07 ± 1.03 (p<0.01) | -6.31 ± 0.71 (p<0.01) | -1.41 ± 0.30 (p<0.01) |
| Hefei | 6.70 ± 0.11 (p<0.01) | -6.73 ± 0.78 (p<0.01) | -0.30 ± 3.43 (p=0.385) |
| Nanjing | 6.50 ± 1.25 (p<0.01) | -11.01 ± 0.90 (p<0.01) | -2.19 ± 0.39 (p<0.01) |
| Ningbo | 3.79 ± 1.16 (p<0.01) | -7.16 ± 0.81 (p<0.01) | -2.51 ± 0.35 (p<0.01) |
| Shanghai | 1.91 ± 1.50 (p=0.204) | -9.91 ± 0.97 (p<0.01) | -4.58 ± 0.43 (p<0.01) |
| Suzhou | 5.84 ± 0.12 (p<0.01) | -7.16 ± 0.81 (p<0.01) | -2.32 ± 0.35 (p<0.01) |




**Table 5.** Correlations of monthly averaged observations against each meteorological parameter from
2005 to 2020.

| City | Correlations | | | | | | | | | |
|------|------|------|------|------|------|------|------|------|------|------|
| | $T_{2m}$ | $U_{10m}$ | $V_{10m}$ | PBLH | TCC | Rain | SLP | SWGDN | $RH_{2m}$ | TROPH |
| Hangzhou | -0.81 | -0.11 | -0.40 | -0.43 | -0.63 | -0.34 | 0.84 | -0.51 | -0.78 | 0.28 |
| Hefei | -0.84 | 0.02 | -0.48 | -0.51 | -0.57 | -0.39 | 0.83 | -0.69 | -0.77 | 0.25 |
| Nanjing | -0.86 | 0.07 | -0.47 | -0.45 | -0.56 | -0.59 | 0.86 | -0.63 | -0.83 | 0.38 |
| Ningbo | -0.84 | 0.39 | -0.71 | -0.14 | -0.70 | -0.47 | 0.86 | -0.54 | -0.82 | 0.07 |
| Shanghai | -0.82 | 0.59 | -0.65 | 0.08 | -0.66 | -0.45 | 0.83 | -0.56 | -0.83 | 0.32 |
| Suzhou | -0.87 | 0.35 | -0.59 | -0.60 | -0.67 | -0.59 | 0.87 | -0.72 | -0.82 | 0.45 |





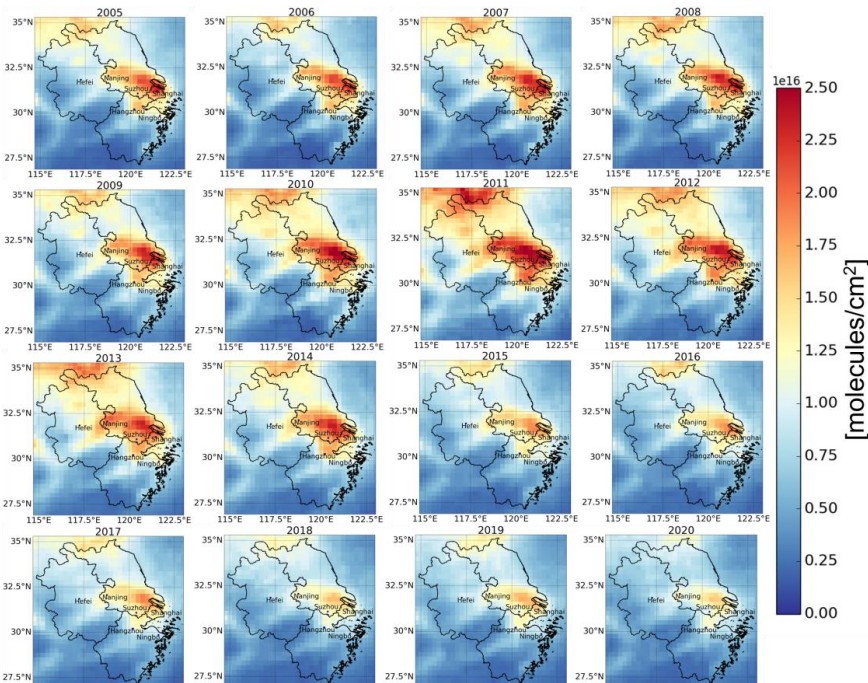

**Figure 1.** Temporal-spatial variabilities of tropospheric NO$_2$ VCDs provided by OMI satellite over the YRD from 2005 to 2020. The three provinces (Anhui, Jiangsu, Zhejiang) and six key megacities (Hefei, Nanjing, Suzhou, Shanghai, Hangzhou, Ningbo) are marked.





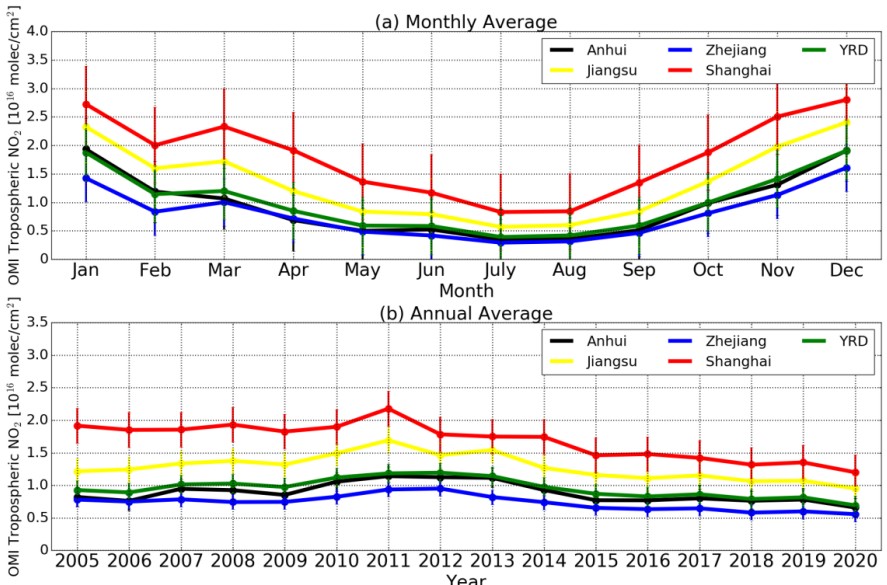

**Figure 2.** (a) Monthly averaged tropospheric NO$_2$ VCDs over the whole YRD region (green dots and lines), Anhui Province (black dots and lines), Zhejiang Province (blue dots and lines), and Jiangsu Province (yellow dots and lines). (b) Same as (a) but for annual average. The vertical error bar is 1σ standard variation (STD) within that month or year.





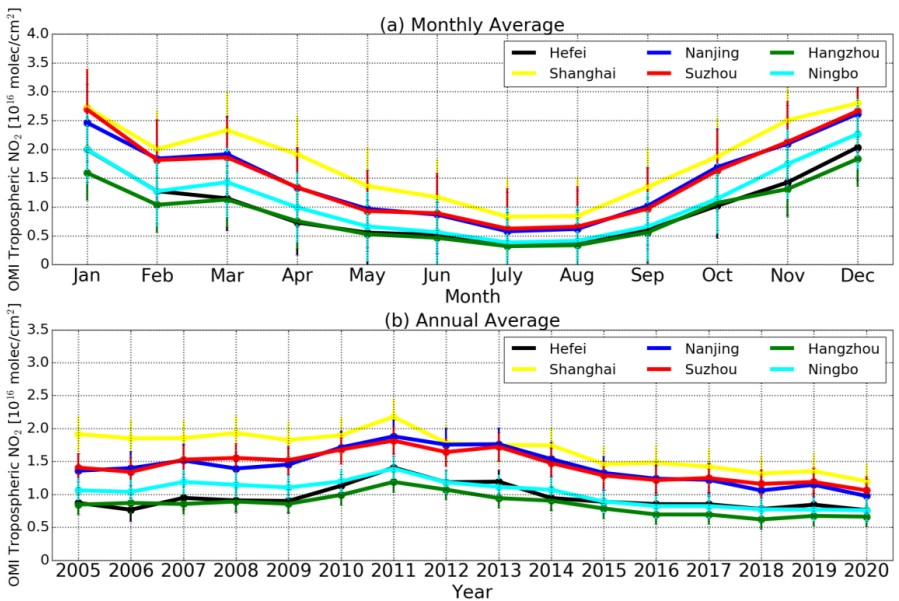

**Figure 3.** (a) Monthly averaged tropospheric $NO_2$ VCDs over Hefei (black dots and lines), Nanjing (blue dots and lines), Shanghai (yellow dots and lines), Suzhou (red dots and lines), Hangzhou (green dots and lines), and Ningbo (cyan dots and lines). (b) Same as (a) but for annual average. The vertical error bar is $1\sigma$ standard variation within that month or year.

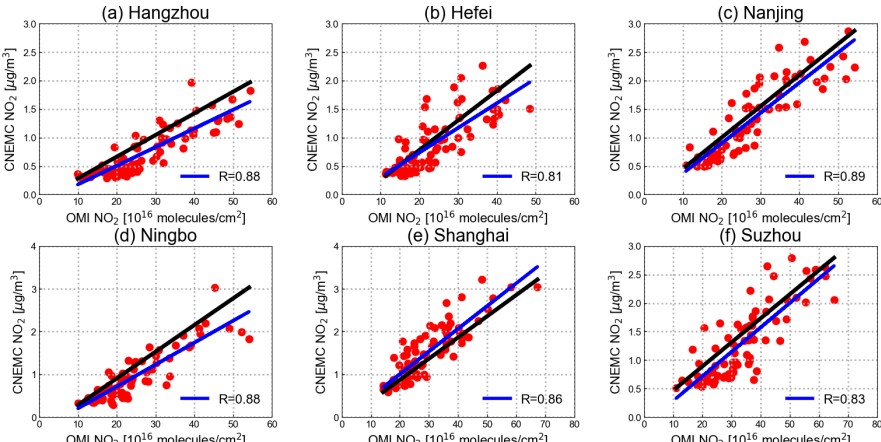

**Figure 4.** Correlation of OMI tropospheric NO$_2$ VCDs against ground-level observations data over Hefei, Nanjing, Shanghai, Suzhou, Hangzhou and Ningbo. We fitted both datasets directly without uniform their units, which does not affect the investigation with respect to the agreement of the two datasets in terms of variabilities. Blue lines are linear fitted lines and black lines are one to one line.





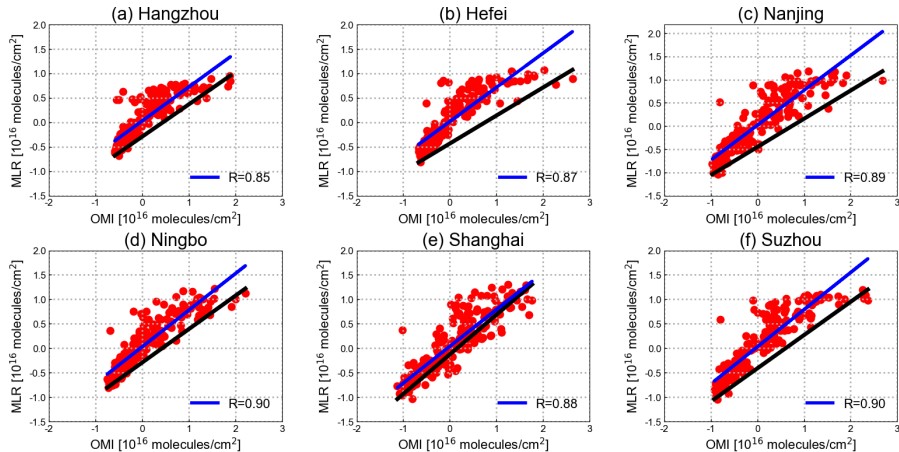

Figure 5. Correlations of OMI tropospheric NO$_2$ VCDs against the MLR model results over Hefei, Nanjing, Shanghai, Suzhou, Hangzhou, and Ningbo. Blue lines are linear fitted lines and black lines are one to one line.





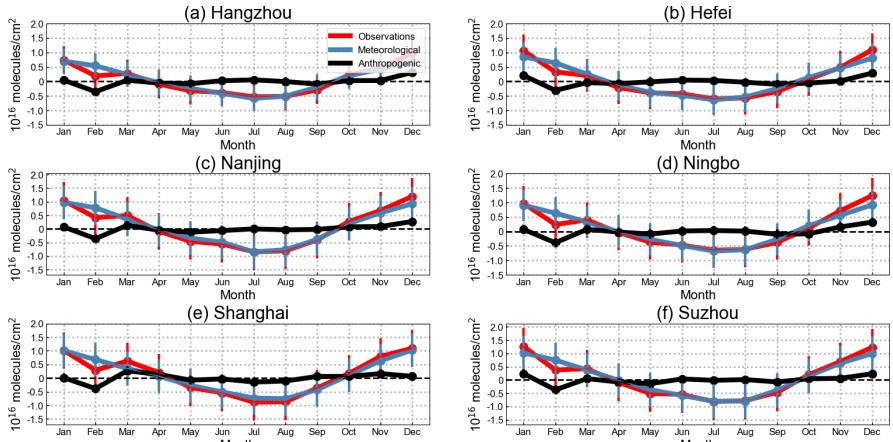

**Figure 6.** Monthly averaged tropospheric NO$_2$ VCDs (red dots and lines) along with the meteorological-driven portions (blue dots and lines) and the anthropogenic-driven portions (black dots and lines) over each city within the YRD. The vertical error bar is 1σ standard variation (STD) within that month.





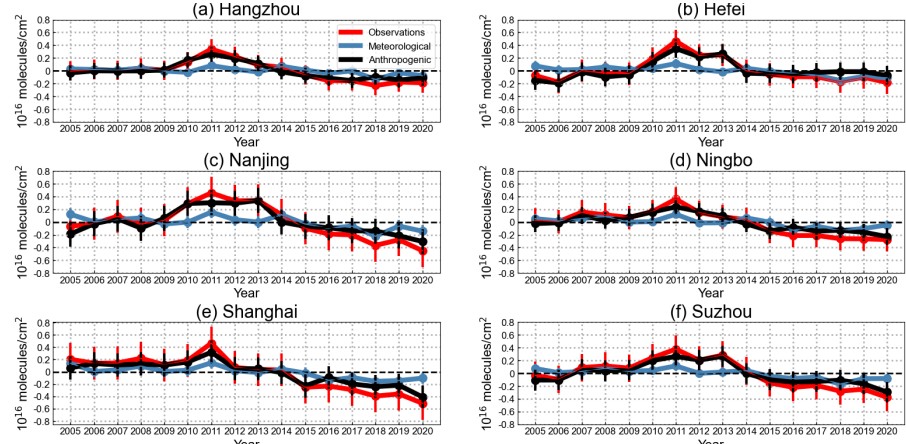


**Figure 7.** The same as Figure 6 but for annual average.







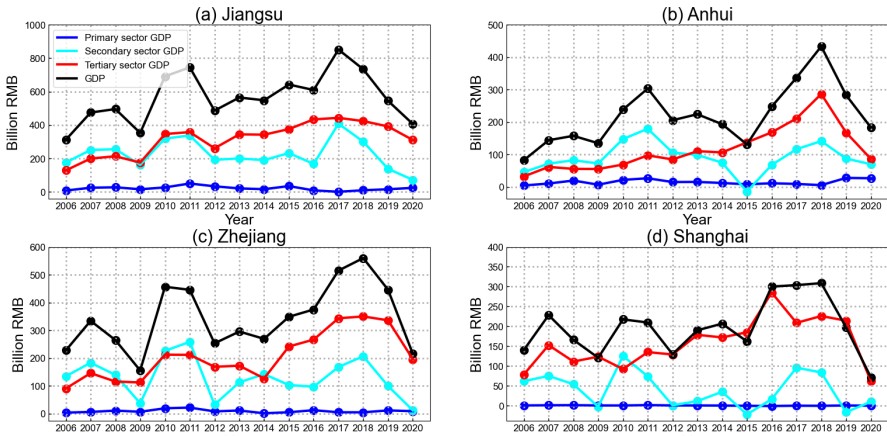


**Figure 8.** Time series of year-to-year increment in GDP, i.e., the increase in GDP at a given year relative to its previous year, over Jiangsu Province, Anhui Province, Zhejiang Province, and Shanghai within the YRD from 2006 to 2020. GDP for total, primary sector, secondary sector, and tertiary sector are marked with black, blue, cyan, and red dots/lines, respectively.




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
