# Peer review of "Space borne tropospheric nitrogen dioxide (NO2) observations 1 from 2005-2020 over the Yangtze River Delta (YRD), China: 2 variabilities, implications, and drivers 3"

_Atmospheric Chemistry and Physics, 2021_

## Author Response (AR1)

**Point-by-point response letter**

**Note:** This file includes comments from the two referees, the corresponding point-by-point responses, and the related changes in the manuscript. The black font are comments from the referees, and the red font are authors' responses as well as the related change clarifications.

**(1) Detailed response to comments from referee #1:**

Thanks very much for your comments, suggestions, and recommendation with respect to improve this paper. The responses to all your comments are listed below.

This manuscript investigates the long-term variabilities of tropospheric $NO_2$ VCDs from 2005-2020 using OMI satellite data product over the Yangtze River Delta (YRD), one of the most densely populated and highly industrialized city clusters in China. The authors also quantify the contributions of meteorology and anthropogenic emissions to the long-term variabilities of tropospheric $NO_2$ VCDs in the major megacities over YRD. They conclude that the inter-annual variabilities of tropospheric $NO_2$ VCDs from 2005 to 2020 over the YRD can be divided into two stages, i.e., an overall increasing trend from 2005 to 2011 and an overall decreasing trend from 2011 to 2020. This is an interesting study and the majority of the work are creative. This manuscript is well written, structured and analyzed convincingly, and its topic fits well within the scope of ACP. I think that this manuscript can improve our knowledge with respect to long-term evolutions of economic and social development, anthropogenic emission over the YRD, and could be of interest to the general atmospheric science community. Minor revisions are recommended.

**Response:** All your comments listed below have been addressed. Please check the point by point response as follows.

**General comments:**

**Comment [1]:** Please provide more detailed descriptions about the YRD region, such as the population, number of cities, area of the region, etc, over Zhejiang, Anhui and Jiangsu Province. This would help the reader understand the reason that YRD region is one of the most densely populated and highly industrialized city clusters in China.

**Response:** Thanks for your suggestion. In our revised version, we have added content to Table S1 (namely Table1 in this response letter) including the detailed descriptions of each province over the YRD region. Please check the marked up file for details.

**Table 1.** Geolocation, the number of measurement site, and population for the 6 megacities within the YRD. Population statistics are based on the seventh nationwide population census in 2020 provided by National Bureau of Statistics of China.

| City | Latitude | Longitude | Number of sites | Altitude (m) | Population (million) |
|------|----------|-----------|-----------------|--------------|----------------------|
| Hangzhou | 30.29 | 120.15 | 11 | 41.7 | 1.19 |
| Hefei | 31.85 | 117.25 | 10 | 29.8 | 0.94 |
| Ningbo | 29.87 | 121.55 | 9 | 5.1 | 0.94 |
| Nanjing | 32.04 | 118.77 | 9 | 8.9 | 0.93 |
| Shanghai | 31.23 | 121.47 | 10 | 4.5 | 2.49 |
| Suzhou | 31.30 | 120.62 | 8 | 3.5 | 1.28 |

**Comment [2]:** In section 3.3 and Figure 5, the authors state that "For each city, the CNMEC ground level $NO_2$ measurements nearest to the OMI ground grid were included for comparison." In each city, there are a number of CNMEC ground stations. However, the authors present comparisons of satellite and ground-based observations for each city. Whether the authors averaged all observed values or some other method? The authors should state the detailed of this process in section 6.

**Response:** Thanks for your suggestion. In this study, ground level $NO_2$ concentrations were taken as the average of all CNMEC stations in each city. The $NO_2$ $VCD_{trop}$ values were taken as the average of all OMI observed grids within the scope of each city. We have modified the corresponding content in Line 330-332, section 3.3. Please check the marked up file for details.

**Detailed comments:**

**Comment [1]:** The phrase "tropospheric $NO_2$ VCDs" is too cumbersome in this manuscript, and please change it to a simple symbol, such as "Tro_$NO_2$" or similar.

**Response:** Done, we changed " tropospheric $NO_2$ VCDs" to "$NO_2$ $VCD_{trop}$". Please check the marked up file for details.

**Comment [2]:** Please include a figure about the monthly average of surface temperature in main cities over YRD in the supplement, same as figure S6.

**Response:** Thanks for your suggestion. We have added Figure S9 (namely Figure 1 in this response letter) including the monthly average of surface temperature in main cities over YRD in the supplement. Please check the marked up file for details.

[Figure]

**Figure 1.** The monthly average of surface temperature in main cities over YRD during 2005-2020.

**Comment [3]:** Line 37, "... inter annual ..." should be "... inter-annual ...".
**Response:** Done. Please check the marked up file for details.

**Comment [4]:** Line 41, "… which cause …" should be "… which causes …".
**Response:** Done. Please check the marked up file for details.

**Comment [5]:** Line 58, "… biogeochemical reaction …" should be "… the biogeochemical reactions …"
**Response:** Done. Please check the marked up file for details.

**Comment [6]:** Line 85, "the rapid developing regions" should be "the rapidly developing regions".
**Response:** Done. Please check the marked up file for details.

**Comment [7]:** Line 102, "polices" should be "policies".
**Response:** Done. Please check the marked up file for details.

**Comment [8]:** Line 259, "Except ..." should be "Except for ...".
**Response:** Done. Please check the marked up file for details.

**Comment [9]:** Line 346, "agreements" should be "agreement".
**Response:** Done. Please check the marked up file for details.

**Comment [10]:** Line 352, "characterized" should be "is characterized"
**Response:** Done. Please check the marked up file for details.

**Comment [11]:** Line 395, "bellow" should be "below".

**Response:** Done. Please check the marked up file for details.

**Comment [12]:** Line 465, "relative" should be "relatively".

**Response:** Done. Please check the marked up file for details.

**Comment [13]:** The usage of "emission" and "emissions" is sometimes misleading, please use it consistently.

**Response:** Done. Please check the marked up file for details.

**Comment [14]:** Please add the units for each meteorological parameter in table 2.

**Response:** Done. Please check Table 2 (namely Table 2 in this response letter) in revised version.

**Table 2.** Meteorological parameters used in the MLR model.

| Parameters | Description | Unit |
|------------|-------------|------|
| T2m | 2m air temperature | °C |
| U10m | 10m zonal wind | m/s |
| V10m | 10m meridional wind | m/s |
| PBLH | Planetary boundary layer height | m |
| TCC | Total cloud area fraction | unitless |
| Rain | Rainfall | kg·m2/s |
| SLP | Sea level pressure | Pa |
| SWGDN | Surface incoming shortwave flux | W/m2 |
| RH2m | 2m Relative humidity | % |
| TROPH | Tropospheric layer Height | m |

**Comment [15]:** I cannot list all technical errors as above. I thus suggest that the authors should check all grammatical errors throughout the manuscript and correct accordingly.

**Response:** Done. Please check the marked up file for details.

**Comment [16]:** Please check the format of each reference and make sure it follows the ACP format.

**Response:** Done. Please check the marked up file for details.

**(2) Detailed response to comments from referee #2:**

Thanks very much for your comments, suggestions, and recommendation with respect to improve this paper. The responses to all your comments are listed below.

Yin et al., present a comprehensive study to look insight to the $NO_x$ trend from 2005 to 2020 over the Yangtze River Delta China by using the OMI space borne observations. Observations revealed that the $NO_x$ experienced an upward and downward trend during 2005-2020, with a threshold of 2011. And they applied the multiple linear regression model to understand the role of anthropogenic emissions and meteorological factor in $NO_x$ level. Model results showed that the seasonal change is mainly attributed to meteorological factor and the long-term trend of $NO_x$ is attributed to emissions. Overall, the dataset and analysis make sense and the topic is with the scope of ACP, I only have some minor comments to be addressed.

**Response:** All your comments listed below have been addressed. Please check the point by point response as follows.

**General comments:**

**Comment [1]:** Section 4.1, could you please provide more information about which two or three meteorological factors influence the level of $NO_x$ more significantly and conduct more discussions about the reasons in the main text?

**Response:** Thanks for your suggestions. We have provided more information about which two or three meteorological factors influence the level of $NO_x$ more significantly and conduct more discussions about the reasons in the main text. As shown in Figure S2, the vast majorities of meteorological contributions over all megacities are from temperature and additional minor contributions over some cities such as Nanjing, Shanghai, and Suzhou are attributed to relative humidity, pressure, or surface incoming shortwave flux (SWGDN) (Agudelo–Castaneda et al., 2014;Parra et al., 2009). Significant negative correlations between temperature and $NO_2$ $VCD_{trop}$ are observed in all megacities (Figure S3, Table 5). Higher temperature tends to decrease $NO_2$ $VCD_{trop}$ and vice versa. This is because higher temperature conditions could accelerate the chemical reaction that destructs $NO_2$ in the troposphere (Pearce et al., 2011;Yin et al., 2021). In addition, surface pressure shows high positive and both surface relative humidity and SWGDN show negative correlations with $NO_2$ $VCD_{trop}$, but their contribution levels are much lower than the temperature. All other meteorological variables only have weak correlations with $NO_2$ $VCD_{trop}$ (Table 5) . Please check the marked up file for details.

**Comment [2]:** Inspired by the text in Line 422-423, I suggest the authors supply two figures (same as Figure 2 and 3 but from 2011-2019) in SI to take a look at the influence

of COVID-19 to the $NO_x$ trend from 2011-2020.

**Response:** Thanks for your suggestions. We have added this two figures in supplement information (Figure S5 and Figure S6, namely Figure 2 and Figure 3 in this response letter). We also obtained the same conclusion as that from Figure 6, indicating the drivers of seasonal cycles of $NO_2$ $VCD_{trop}$ deduced above are consistent over years. Please check the marked up file for details.

[Figure]

**Figure 2.** (a) Monthly averaged tropospheric $NO_2$ VCDs over the whole YRD region (green dots and lines), Anhui Province (black dots and lines), Zhejiang Province (blue dots and lines), and Jiangsu Province (yellow dots and lines) during 2011 to 2019. (b) Same as (a) but for annual average. The vertical error bar is $1\sigma$ standard variation (STD) within that month or year.

[Figure]

**Figure 3.** (a) Annual averaged tropospheric NO₂ VCDs over the whole YRD region (green dots and lines), Anhui Province (black dots and lines), Zhejiang Province (blue dots and lines), and Jiangsu Province (yellow dots and lines) during 2011 to 2019. (b) Same as (a) but for annual average. The vertical error bar is 1σ standard variation (STD) within that month or year.

**Comment [3]:** I can understand the motivation of using the GDP data in the discussions, while it seems that the GDP cannot be a perfect explanation for the trend of NOₓ emission, so I believe Figure 8 is not so important and can be moved to SI. By the way, I encourage the authors to collect some information about the motor vehicle emissions and major industrial emissions data in this region and analysis the NOₓ trend with these emissions.

**Response:** Thanks for your suggestions. We have moved Figure 8 to supplement information (Figure S11). In Figure S12 (namely figure 4 in this response letter), we

further analyzed the variabilities of $NO_x$ emissions over the YRD region from 2008 to 2017 by category provided by the MEIC inventory, including motor vehicle emissions, major industrial emissions, resident emissions and power emissions (http://meicmodel.org, last accessed: February 25, 2022) (Li et al., 2017;Zheng et al., 2018). The results show that the decreases in $NO_2$ $VCD_{trop}$ over the YRD during 2011 to 2013 are attributed to the reductions of industrial and power emissions, during 2013 to 2014 are mainly attributed to the reductions of motor vehicle emissions and power emissions, and after 2014 are attributed to the reductions of motor vehicle emissions, power emissions and industrial emissions. We have added the corresponding content to line 473-481. Please check the marked up file for details.

[Figure]

**Figure 4.** Each categories $NO_x$ emissions including motor vehicle emissions, major industrial emissions, resident emissions and power emissions from 2008 to 2017 which are extracted by MEIC inventory (http://meicmodel.org, last accessed: February 25, 2022) over YRD region.

**Comment [4]:** Line 271-273, please check the decrement of Anhui and the total YRD, as the data values are the same, may be a typo.

**Response:** Thanks for your reminder. We have revised this mistake in the correspond content. The total decrements over Anhui in 2020 relative to 2011 is $(4.82 \pm 0.35) \times 10^{15}$ molecule/m$^2$ $(43.26 \pm 3.07)$ % (Line 266-267). Please check the marked up file for details.

**Comment [5]:** Figure S2, $NO_2$ change to subscript.

**Response:** Thanks for your reminder. We have modified the figure in supplement. Please check the marked up file for details.

**Reference**

Agudelo–Castaneda, D. M., Calesso Teixeira, E., and Norte Pereira, F.: Time–series analysis of surface ozone and nitrogen oxides concentrations in an urban area at Brazil, Atmospheric Pollution Research, 5, 411-420, https://doi.org/10.5094/APR.2014.048, 2014.

Li, M., Liu, H., Geng, G., Hong, C., Liu, F., Song, Y., Tong, D., Zheng, B., Cui, H., Man, H., Zhang, Q., and He, K.: Anthropogenic emission inventories in China: a review, Natl Sci Rev, 4, 834-866, 10.1093/nsr/nwx150, 2017.

Parra, M. A., Elustondo, D., Bermejo, R., and Santamaría, J. M.: Ambient air levels of volatile organic compounds (VOC) and nitrogen dioxide ($NO_2$) in a medium size city in Northern Spain, Sci Total Environ, 407, 999-1009, https://doi.org/10.1016/j.scitotenv.2008.10.032, 2009.

Pearce, J. L., Beringer, J., Nicholls, N., Hyndman, R. J., and Tapper, N. J.: Quantifying the influence of local meteorology on air quality using generalized additive models, Atmos Environ, 45, 1328-1336, https://doi.org/10.1016/j.atmosenv.2010.11.051, 2011.

Yin, H., Liu, C., Hu, Q., Liu, T., Wang, S., Gao, M., Xu, S., Zhang, C., and Su, W.: Opposite impact of emission reduction during the COVID-19 lockdown period on the surface concentrations of $PM_{2.5}$ and $O_3$ in Wuhan, China, Environmental Pollution, 289, 117899, https://doi.org/10.1016/j.envpol.2021.117899, 2021.

Zheng, B., Tong, D., Li, M., Liu, F., Hong, C., Geng, G., Li, H., Li, X., Peng, L., Qi, J., Yan, L., Zhang, Y., Zhao, H., Zheng, Y., He, K., and Zhang, Q.: Trends in China's anthropogenic emissions since 2010 as the consequence of clean air actions, Atmos. Chem. Phys., 18, 14095-14111, 10.5194/acp-18-14095-2018, 2018.